# Optimal age targeting for pneumococcal vaccination in older adults; a modelling study

Deus Thindwa [1,2,3] ✉, Samuel Clifford[1,2], Jackie Kleynhans [4,5], Anne von Gottberg [4,6], Sibongile Walaza[4], Susan Meiring [4], Todd D. Swarthout [3,7,8], Elizabeth Miller[2], Peter McIntyre[9], Nick Andrews[10], Zahin Amin-Chowdhury[10], Norman Fry [10], Kondwani C. Jambo [3,11], Neil French[3,12], Samanta Cristine Grassi Almeida[13], Shamez N. Ladhani[10], Robert S. Heyderman [3,7], Cheryl Cohen [4,5], Maria Cristina de Cunto Brandileone[13] & Stefan Flasche [1,2]

Invasive pneumococcal disease (IPD) risk increases with age for older adults whereas the population size benefiting from pneumococcal vaccines and robustness of immunogenic response to vaccination decline. We estimate how demographics, vaccine efficacy/effectiveness (VE), and waning VE impact on optimal age for a single-dose pneumococcal vaccination. Age- and vaccine-serotype-specific IPD cases from routine surveillance of adults ≥ 55 years old (y), ≥ 4-years after infant-pneumococcal vaccine introduction and before 2020, and VE data from prior studies were used to estimate IPD incidence and waning VE which were then combined in a cohort model of vaccine impact. In Brazil, Malawi, South Africa and England 51, 51, 54 and 39% of adults older than 55 y were younger than 65 years old, with a smaller share of annual IPD cases reported among < 65 years old in England (4,657; 20%) than Brazil (186; 45%), Malawi (4; 63%), or South Africa (134, 48%). Vaccination at 55 years in Brazil, Malawi, and South Africa, and at 70 years in England had the greatest potential for IPD prevention. Here, we show that in low/middle-income countries, pneumococcal vaccines may prevent a substantial proportion of residual IPD burden if administered earlier in adulthood than is typical in high-income countries.

*Streptococcus pneumoniae* (pneumococcus) is a major global cause of childhood mortality[1,2], but also causes a high burden of disease among older adults[2,3]. Two vaccines have been used to prevent pneumococcal disease in older adults ≥55 years-old (y): a 13-valent pneumococcal conjugate vaccine (PCV13) and a 23-valent pneumococcal polysaccharide vaccine (PPV23)[4]. Recently, 15- and 20-valent PCVs (PCV15, PCV20) have also been licensed and recommended for older adults in the United States[5,6].

Although routine infant PCV programmes have generated indirect protection against vaccine-serotype (VT) pneumococcal disease among older adults[7,8], a substantial disease burden remains, composed of serotypes not targeted by childhood PCV programmes and residual circulation of VT[2,9]. Among high-income countries (HICs) with a mature infant-PCV13 programme, PCV13- and PPV23-targeted serotypes caused about 15 and 42%, respectively, of invasive pneumococcal disease (IPD) in older adults[2,10]. Routine infant-PCV programmes

in many low- and middle-income countries (LICs and MICs) have often led to less pronounced herd effects with continued circulation of VTs especially in the unvaccinated adult population[11,12], who happen to have no access to routine pneumococcal vaccination[13].

The recommended age for pneumococcal vaccination in older adults in HICs is typically either at 60 years or 65 years[4,14]. However, only a single study has assessed the age at which the most gain from such a programme is seen (in the Australian context)[15]. As the risk of severe pneumococcal disease increases with age[10,15], waning of protection from vaccination early in older adulthood risks disease at ages of highest disease incidence, whereas vaccination late in older adulthood cannot address the substantial disease burden among a large pool of susceptible older adults who have not yet been vaccinated.

In this modelling study, we explore the optimal age-targeting for a single-dose pneumococcal vaccination against VT-IPD in older adults living without human immunodeficiency virus (HIV) in Brazil, England, Malawi, and South Africa.

## Results

### Population demographics and IPD burden
Of the 27.7 million (m) older adults (≥ 55 years) in Brazil, 16.5 m in England, 66,589 in Blantyre Malawi, and 6.9 m in South Africa, the proportion of older adults aged 55 years to <65 years was 51.3%, 51.0% and 53.8% in Brazil, Malawi and South Africa, substantially higher than the 39.1% in England. During the study period, Brazil (2015–2017), England (2016–2019), Malawi (2016–2019) and South Africa (2015–2018) reported 559, 13,971, 19, and 537 IPD cases in adults ≥ 55 years equivalent to an average annual number of cases of 186.3, 4,658·0, 4.8 and 134.3, respectively. Of these in the ≥ 55 years, 44.5%, 19.9%, 62.5%, and 47.9% were in < 65 years olds. Cases caused by PCV13 serotypes accounted for 61.4% (343/559), 21.0% (2,936/13,971), 41.0% (8/19), and 32.8% (176/537) of all IPD in Brazil, England, Malawi and South Africa, and PPV23 serotypes for 97.9% (547/559),

72.5% (10,126/13,971), 94.8% (18/19), and 73.2% (393/537), respectively (Fig. 1).

The exponential model fitted the increase in IPD incidence with age well. The estimated IPD incidence in 85 y was higher than in 55 years olds by 2·48-fold (95% confidence interval [95%CI]: 2.13–2.83) in Brazil, 2.19-fold (95% CI: 0.14–4.51) in Malawi, and 2.25-fold (95% CI: 1.88–2.62) in South Africa. In England the incidence increased more steeply to 11.00-fold (95% CI: 10.90–11.40) higher in 85 years than 55 years (Fig. 1).

While the estimated number of IPD cases declined with age in Brazil, Malawi and South Africa, it increased in England. On the other hand, age-specific IPD incidence of total IPD incidence (age-scaled IPD incidence) increased with age in all settings irrespective of serotype, and with high uncertainty in Malawi due to small case numbers (Fig. S4, Fig. S5).

### Optimal age for vaccination
The optimal age for vaccination, the age of a single-dose vaccination that could prevent most IPD cases, is attained when vaccines are given earlier in the considered age range in the LIC/MIC setting, but not for England (Fig. 2). In the base scenario with rapid waning of PPV23 VE, we found that the highest proportion of all IPD cases could be preventable if adults aged 55 years are vaccinated in Brazil (30.5%, 22.4–43.3), in Malawi (31.0%, 7.7–96.6), and in South Africa (20.0%, 13.7–28.7), and if adults aged 70 years are vaccinated in England (13.7%, 10.8–17.4) compared to a scenario without vaccination (Table S1). Also, in the base scenario with rapid waning VE, a higher proportion of all preventable cases are estimated for using PCV20 vs PPV23 relative to a scenario without vaccination in Brazil (61.3%, 42.7–89.2) vs (30.1%, 20.6–43.5), Malawi (64.5 %, 13.0–92.2) vs (28.5%, 5.7–73.3), and South Africa (53.1%, 39.7–73.2) vs (19.8%, 13.0–29.2) among adults aged 55 years, and in England (27.2%, 21.6–34.5) vs (13.6%, 10.4–17.6) among adults aged 70 years,

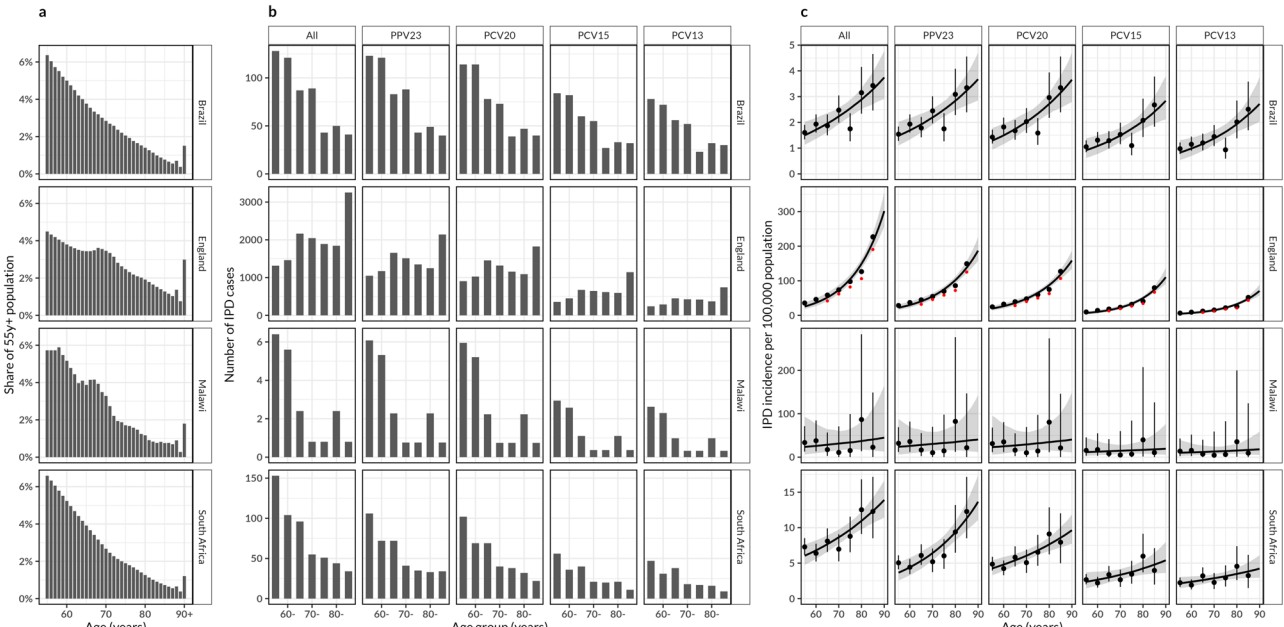

**Fig. 1 | Population demographics and invasive pneumococcal disease (IPD) burden. a** Among individuals who are aged ≥ 55 years, the proportion/share in annual age groups in Brazil, England, Malawi and South Africa as estimated from their national censuses, based on five years rolling average smoothed population counts to control for demographic stochasticity. **b** Number of IPD cases in five year age bands in older adults stratified by serotype in Brazil (2015–2017), England (2016–2019), Blantyre Malawi (2016–2019) and South Africa (2015–2018), reported from at least four years post-infant PCV introduction in each country,

**c** Serotype-specific reported and predicted IPD incidence per 100,000 population between 55 years and 90 years in Brazil, England, Malawi and South Africa. The black circle represents estimated IPD cases per 100,000 population, the vertical line through the circle represents a 95% uncertainty interval in estimated IPD case number, the curve line is the exponential model fit and the ribbon represents a bootstrapped 95% confidence interval for the fitted line. The red and black points for England represent estimated IPD cases in the presence and absence of PPV23 vaccination, respectively.

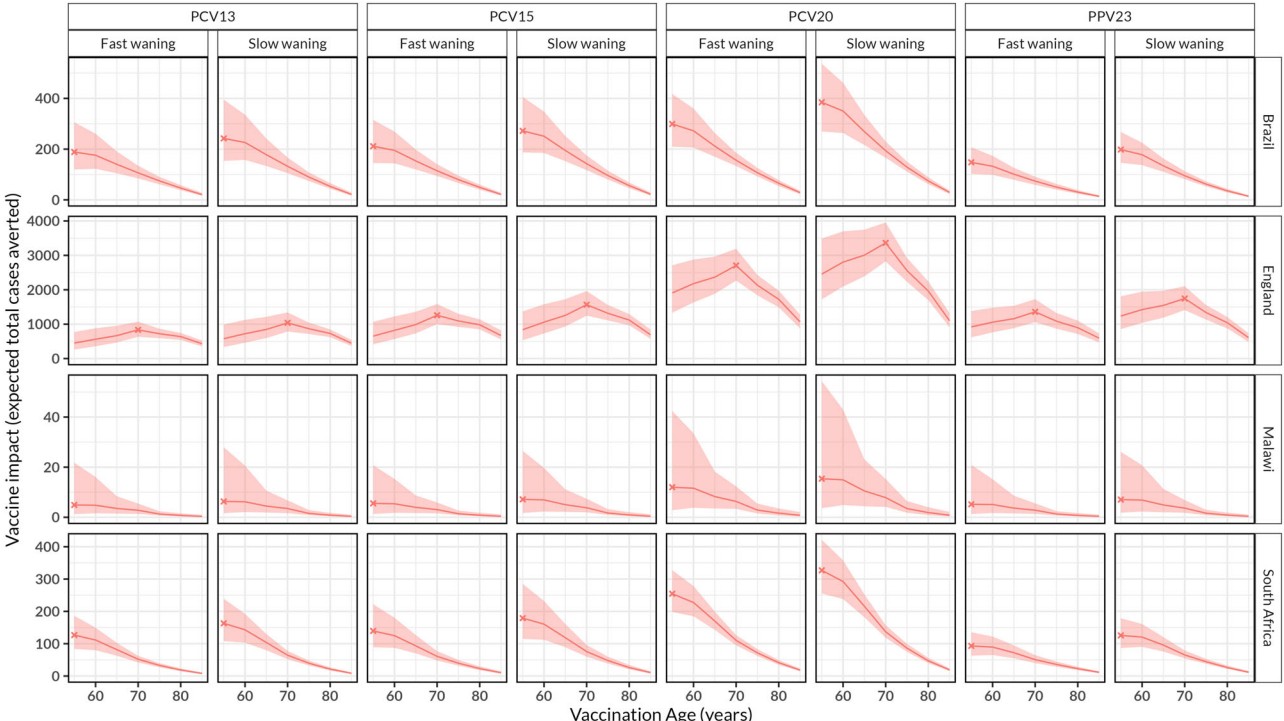

**Fig. 2 | The impact of routine pneumococcal vaccination in older adults aged ≥ 55 years old (y).** The expected absolute number of total IPD cases averted for the rest of age cohort lifetime by vaccinating every older adult in the age cohort stratified by country and vaccine product, under the scenario of age-independent initial vaccine efficacy/effectiveness (VE) and waning VE in Brazil, England, Malawi and South Africa. The red lines represent cohort model mean estimates and the shaded red ribbon represents 95% bootstrap confidence intervals for the mean estimates. The X corresponds to the optimal age for pneumococcal vaccination. In Brazil, Malawi and South Africa, most cases are preventable at age 55 years whereas in England this is achieved at age 70 years.

respectively (Table S2). Furthermore, a higher proportion of IPD cases are preventable under slow vs rapid waning of PPV23 VE among adults aged 55 years in Brazil (40.7%, 29.0–57.1) vs (30.1%, 20.6–43.5), Malawi (38.6 %, 8.0–77.2) vs (28.5%, 5.7–73.3), South Africa (26.4%, 18.2–38.3) vs (19.8%, 13.0–29.2), and among adults aged 70 years in England (17.5%, 13.8–22.0) vs (13.6%, 10.4–17.6), respectively (Table S3).

The optimal age for vaccinating the lowest number of individuals with single dose pneumococcal vaccine to prevent a reported case of IPD (vaccination efficiency) is estimated to differ by country, vaccine product or waning VE assumption (Fig. 3). In a base case scenario with rapid waning of PPV23 VE, optimal age for vaccination efficiency is achieved through vaccination at 60 y in Malawi 684 (244–2,009), 65 years in Brazil 10,283 (8,165–13,162), and 75 years in South Africa 3,643 (3,004–4,555) and at 80 years in England 352 (287–438) (Table S4).

**Sensitivity analyses**

If a scenario of age-dependent initial VE is considered in a sensitivity analysis, vaccine impact (total cases averted) remains highest at 55 years in Brazil, Malawi, and South Africa irrespective of vaccine product and assumption of waning VE as was the case with the base case scenario of age-independent initial VE. In contrast, optimal age for vaccination drops to 60 years in England (Fig. S6). In this scenario, the optimal age of vaccination efficiency is achieved by vaccination of 60 years in Brazil, Malawi and South Africa, irrespective of vaccine product or assumptions of waning VE. In England, efficiency is achieved in 60 years age cohort for PCV20 and PPV23 irrespective of waning VE assumption, and in 80 years or 85 years for PCV13's and PCV15's slow or fast waning VE, respectively (Fig. S7).

## Discussion

We have assessed the optimal age-targeting for a single-dose PCV/PPV vaccination against VT-IPD in older adults ≥ 55 years in Brazil, England, Malawi, and South Africa and find that vaccinating at 55 years in considered LIC/MICs maximises preventable burden of IPD while vaccination at 70 years is optimal in England. These findings suggest that the optimal age for vaccination may differ between countries and is driven by population age demographic, age-dependent IPD incidence and VE. Our findings were robust across pneumococcal vaccine products, and to alternative assumption on waning of VE.

The increasing incidence of IPD in older adulthood is outweighed by many more individuals in their fifties in LIC/MICs. The different age population structure in England, with a higher proportion of ≥ 55 years in the middle to older adult ages, results in the optimal age for vaccination peaking later at 70 years. The optimal age of vaccination in England also aligns with results from an Australian study that explored the role of timeliness in the cost-effectiveness of older adult vaccination and found that most deaths and hospitalisations were prevented if PCV13 was given to 70 years, a peak age of prevented burden before dropping in remaining older ages[15].

The age with lowest number of individuals needed to vaccinate to prevent a case (optimal age of vaccination efficiency) differed by country, vaccine product or assumption of waning VE, in part reflecting the differences in sensitivity of IPD surveillance and the complex interplay between different factors influencing vaccination efficiency. Although this finding accounts for life expectancy, it does not consider quality-adjusted life-years losses due to IPD which could shift the optimal age of vaccination efficiency downwards because more individuals affected by IPD would be added to the fifties than oldest ages. Although our modelled scenarios assumed 100% vaccination coverage, which may not reflect real-world vaccine uptake, we aimed to

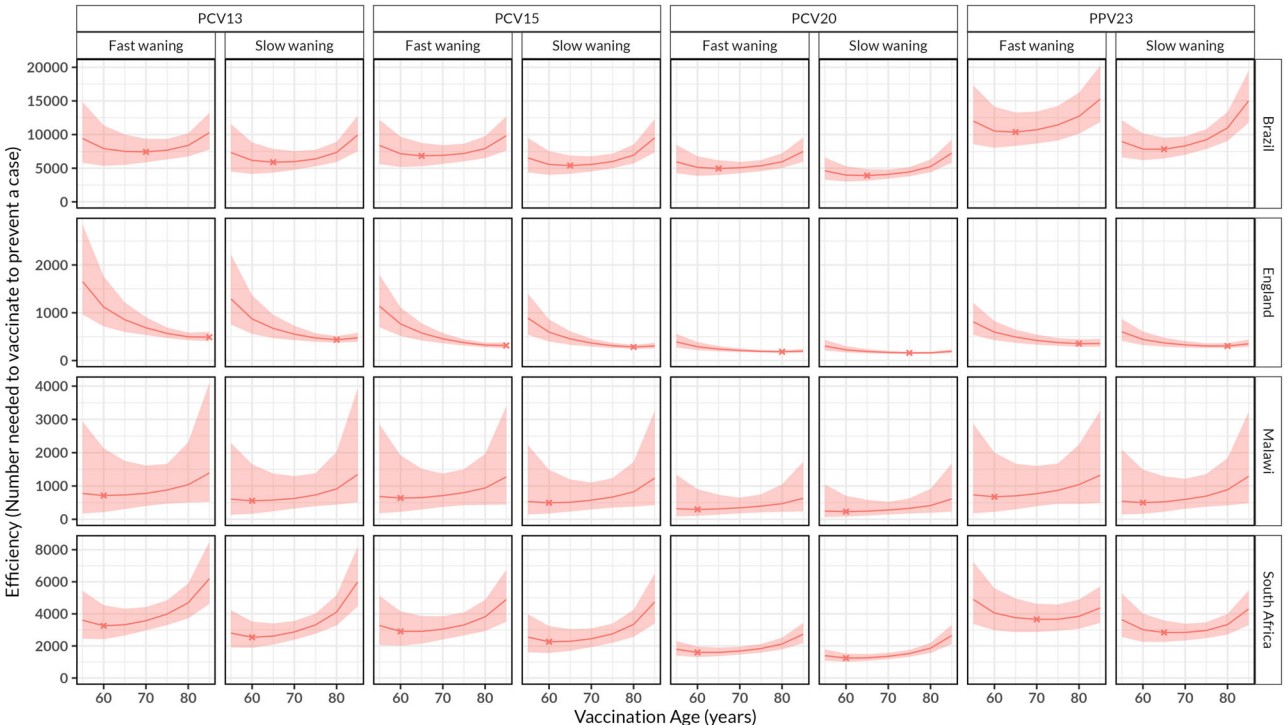

**Fig. 3 | The efficiency of routine pneumococcal vaccination in older adults aged ≥ 55 years old (y).** The number of individuals needed to vaccinate to prevent a case in each age of vaccination, stratified by country and vaccine product, under assumptions of age-independent initial vaccine efficacy/effectiveness (VE) and waning VE in Brazil, England, Malawi and South Africa. The red lines represent cohort model mean estimates and the shaded red ribbon represents 95% bootstrap confidence intervals for the mean estimates. The X represents the optimal age for efficiency of pneumococcal vaccination. Efficiency of vaccination varies by waning VE assumption and country reflecting sensitivity in reported invasive pneumococcal disease cases.

characterise the potentially preventable burden and that the predicted impact will scale linearly with uptake in this model thus not affecting optimal age of vaccination unless uptake is differential by age groups.

Our cohort model prediction of more preventable cases in all settings with the use of PCV20 vs PPV23 under similar conditions reflects higher efficacy/effectiveness of PCV20 than PPV23[2]. Moreover, a larger number of IPD preventable cases by PCV20 than lesser-valent PCVs underlines the importance of additional serotypes 8, 10 A, 11 A, 12 F and 15B and/or 22 F, 33 F covered by PCV20 in the remaining IPD burden in the mature infant-PCV era[10]. Generally, other disease end-points including non-bacteraemic pneumonia may also be important for vaccine choice e.g., PCV13 is shown to be more efficacious in preventing VT non-bacteraemic pneumonia than PPV23 (46% vs 21%)[2], and if included in this analysis, it may likely heighten PCV13 impact to similar levels or higher than PPV23 in England, albeit with higher vaccine cost. With the likely upcoming use of PCV15 and PCV20 in infants in the near future[16], the vaccine preventable fraction of adult disease is likely to diminish correspondingly because of the indirect effects of such infant programme, and thus reducing the benefit of adult programmes that target the same serotypes as those targeted in infants. On the other hand, phase 3 trials of the investigational PCV21 by MERCK are underway after receiving a breakthrough therapy designation from the US Food and Drug Administration for prevention of IPD. PCV21 targets serotypes that account for 85% of IPD in >65 years in the US including 8 serotypes not targeted by currently licensed vaccines[17], and it is expected to improve tackling the remaining burden of pneumococcal disease in older adults, in the long absence of vaccines that target surface proteins common to all serotypes[18]. However, for LICs/MICs, PCV15, PCV20 and PCV21 may likely not be accessible soon due to their high initial costs.

Of note, we excluded older adults with HIV because they typically are considered separately for pneumococcal vaccination because of their high risk of disease before they reach old age[12]. However, inclusion of older adults with HIV may likely elevate IPD burden in early older adulthood because of their huge presence in early than late adulthood due to survival bias thereby re-enforcing vaccinating early in older adulthood, particularly in Malawi and South Africa where adult HIV prevalence is relatively high[19,20]. In this analysis, the contribution of PCVs' indirect protection from older adults to overall vaccine impact is ignored because older adults have low carriage rates unlike children who are the main transmitters, and the use of PPV23 does not protect against carriage and therefore unlikely to generate substantial herd immunity[21]. Sequential dosing of vaccine regimens was not modelled, although PCV15 followed by PPV23 is now recommended by the advisory committee on immunisation practices (ACIP) in the United States[22]. This sequential regimen would likely generate higher impact than a single dose of PCV20 or PPV23, albeit with cost-effectiveness implications that need further research, particularly in LICs[2].

PCVs and PPV23 are reported to have limited efficacy/effectiveness against serotype 3 pneumococcal carriage and disease[23], and our inclusion of serotype 3 IPD cases in this analysis imply that our vaccine impact estimates are likely biased upwards. PPV23 is already in use in ≥ 65 years in England[24], and we show in this analysis that the effects of PPV23 use have reduced the original IPD burden (Fig. 1), and without adjusting for the current PPV23 vaccination programme, our optimal age for vaccination in this country may be overestimated. Reported IPD case data in LMICs are usually incomplete or under-ascertained due to limited resources e.g., only 19 IPD cases were reported in Malawi and even less so when stratified by age and serotype[3]. Thus, our results should be cautiously interpreted. Despite potential biases of case underreporting, it seems reasonable to assume that underreporting is consistent across adult age groups, such that a relative change in IPD incidence by age to identify optimal age-targeting vaccination is less likely affected. It is also worth noting that our study uses demographic

snapshots and that there are ongoing demographic changes, particularly in Africa. Generally, with increasing life expectancy we would expect the optimal age of vaccination to increase, but this is a process spanning decades.

In conclusion, the optimal age-targeting for vaccination is largely driven by population age demographic, age-dependent IPD incidence and VE. In contrast to the typical use in adults in HICs, we find that pneumococcal vaccination in 55 years older adults in LIC/MICs may be the most effective strategy in reducing the IPD burden among older adults without HIV, although affordability, cost-effectiveness and infant vaccination plans will also need to considered when designing a strategy to protect older adults against pneumococcal disease.

## Methods
### Study sites
We included England (HIC), Brazil and South Africa (MICs), and Blantyre in Malawi (LIC) in this analysis, to explore the optimal age-targeting for pneumococcal vaccination, as all have long-standing adult pneumococcal surveillance programmes at the national or subnational level[9,24–26]. More details on the respective IPD surveillance systems are in the appendix (Text S1). Infant-PCV13 has been in use since 2011 in South Africa (two primary doses and booster (2 + 1) schedule) and Malawi (3 + 0 schedule), with estimated 90–95% vaccination coverage[11,26]. In England, routine PCV13 immunisation programme with 92% booster dose coverage has been in place since 2010, initially using a 2 + 1 schedule, switching to 1 + 1 schedule in 2020[27]. In Brazil, a routine infant-PCV10 programme was implemented in 2010 with a 3 + 1 schedule, switched to a 2 + 1 in 2016, with average coverage of 94% between 2015 and 2017[28].

In England, PPV23 has been recommended for risk groups since 1992, and for all adults aged ≥ 65 years since 2003, with single-dose coverage rising to 70% in 2018[24,29]. In Brazil, PPV23 is not included in the national immunisation programme but has been recommended for use since the 1980s, and is available free of charge at the centers of special immunobiological for people above 2 years old including institutionalised older adult, and thus coverage of < 1% among all older adults above 60 years[30]. In South Africa, although adult pneumococcal vaccination is recommended, there is no routine programme nor nationally accepted guidelines[31]. In Malawi, neither routine adult pneumococcal vaccination programme nor national guidelines exist[13].

### Population demographics
The population demographics for adults ≥ 55 years were obtained in annual age strata from population censuses conducted in 2010 for Brazil[32], 2011 England (updated in 2017)[33], 2018 Malawi[34], and 2011 South Africa[35]. To align with IPD surveillance timelines in respective countries, annual age population censuses were projected at a constant growth rate of 0.8% during 2010–2016 in Brazil and 1.3% during 2011–2016 in South Africa[19,36]. For England and Malawi, 2017 and 2018 populations already aligned with IPD surveillance time. Annual age population estimates were smoothed to reduce demographic stochasticity in downstream results by applying a 5-year moving average[37].

### Invasive pneumococcal disease burden
For each site we used data on IPD burden in older adults in the presence of a mature infant-PCV programme, defined as at least 4 years after infant-PCV introduction, to avoid inclusion of ongoing changes in IPD burden attributable to indirect effects from the childhood programme[38]. The number of IPD cases caused by all serotypes, and serotypes targeted by PCV13, PCV15, PCV20 and PPV23 in annual age strata (55 years to 85 years+) were obtained from laboratory-based surveillance in each country (Fig. S1). We calculated age and serotype distribution and proportionally inflated estimates by the number of IPD cases without serotyping available to correct for proportion

serotyped. In England, where PPV23 has been in use in 65 years+ at 70% uptake, we back-inflated the number of IPD cases in order to correct for the fact that in the absence of PPV23 vaccination, IPD incidence would be higher, especially in ages eligible for PPV23 vaccination. Details about back-inflation calculation are in the appendix (Text S2). Age aggregated (55–59 years, 60–64 years, 65–69 years, 70–74 years, 75–79 years, 80–84 years and 85 years+) and serotype-specific IPD incidence was calculated by dividing by the age-group specific population estimates. Due to the small number of reported IPD cases and incomplete serotyping information in Malawi, we only calculated the serotype distribution for all cases and assumed that it was the same in each age group.

The focus of this study was vaccination strategies for older adults living without HIV because adults living with HIV often have independent pneumococcal vaccine recommendations[12]. Thus, for adult HIV prevalence of >10% in South Africa[39], we adjusted IPD incidence estimates for HIV status. For 201 (18.7%) reported IPD cases, HIV status was reported. We took a 30% random subset of IPD cases with known HIV status and estimated a 49.3% proportion of IPD cases without HIV for this subset. We assumed that a similar proportion of IPD cases with unknown HIV status would be without HIV. We further accounted for similar serotype distribution in those with and without HIV. In Malawi, IPD cases could not be stratified by HIV infection status, and thus we used the age-dependent HIV infection rates in the general adult population and applied the relative propensity for IPD in older adults with and without HIV from South Africa to infer the incidence of IPD among older adults without HIV by age.

We modelled the reported age-specific IPD incidence among older adults without HIV per country as a function of age using an exponential growth model. Details about model fitting are in appendix (Text S3). The expected incidence at age $a$ is:

$$E(I|a) = \gamma e^{\beta a}, \tag{1}$$

where $E(I|a)$ is the expected IPD incidence at age $a$, $\beta$ is the growth rate, and $\gamma$ is a constant of proportionality. Uncertainty was obtained by bootstrap sampling 1000 times using the fitted parameter means and covariance matrix and summarisation of uncertainty via 95% quantiles of samples.

### PCV and PPV vaccine efficacy/effectiveness
A previous systematic review identified studies that estimate vaccine efficacy/effectiveness (VE) against IPD from time since vaccination[2]. Four observational studies in HICs estimated VE at two or more time points for PPV23[24,29,40–42]. One randomised controlled trial estimated VE against IPD and community-acquired pneumonia (CAP) following PCV13 use in older adults and found consistent 75 and 46% respective efficacy for 5 years following vaccination[43,44]. Thus, we assumed that PCVs' efficacy against IPD would continue to stay stable for five years, and thereafter decline in the same way as the scenario for PPV23. We represented the VE as a function of time since vaccination using piecewise constant models. We considered waning VE from Andrews et al.[29] and Djennad et al.[24] in the vaccination impact cohort model as they provided the fastest and slowest waning VE, respectively, of the four studies. The piecewise constant functions that define initial VE and waning VE are shown in supplementary (Figs. S2, S3). The VE was sampled using bootstrap sampling from a normal distribution centred on the mean VE at that time and standard deviation derived from the reported 95% interval.

Little evidence is available on changes in PPV23 or PCV13 VE by age at administration[15,45]. A UK study found no significant difference in VE of PPV23 when administered to 65 years, 75 years or 85 years adults, although point estimates suggested a decline with age[24]. There is also some evidence that immune response to immunisation is partially impaired later in life[2]. Thus, as a base case we assumed that VE for all

formulations was independent of the age of administration, but included a sensitivity analysis assuming that VE was 33% less of initial VE if given to adults aged 65–74 years and 44% less of initial VE if given to adults aged 75 + years using point estimates of VE of PPV23 against IPD in the 5 years following administration in adults <65 years of 54%, decreasing to 36% for 65–74 years and 30% for 75 years + [24].

## PCV and PPV vaccine impact model
We developed a cohort model to simulate the risk for IPD in adults > 55 years in all four countries. Vaccine impact in the cohort model was a function of 1) smoothed population counts, 2) fitted IPD incidence using exponential growth model, and 3) fitted vaccine VE using piecewise constant model, stratified by age, country, vaccine product, age-dependency on initial VE, and waning VE assumption. Time steps in the model were set at 1 year. Age-dependent all-cause mortality was set to match observed population demographics of older adults living without HIV in all countries, and age-dependent IPD risks were set to match fitted IPD incidence in respective countries and years. We calculated the vaccine preventable number of IPD cases as the lifetime number of cases averted through vaccinating 100% of adults at the specified age for vaccination under the scenarios of age-dependency on initial VE and waning VE. In particular, we calculated the percentage of preventable cases as the proportion of IPD preventable in a lifetime, with denominator being all IPD that can occur in a cohort of adults ≥55 years, and the numerator as the number of IPD cases that can be prevented if vaccinated at a certain age cohort. We estimated the efficiency of the alternative strategies by reporting the number of age cohort individuals needed to vaccinate (or number of doses needed to administer to age cohort) in order to prevent a case.

Sensitivity analysis assessed the influence of age-dependent initial VE on vaccine impact. All the datasets used in the study were collected and shared via Microsoft Excel 2016 and Microsoft Access 2016. All statistical analyses were conducted using R language v4.1.1

## Ethical approval
In Brazil, no patient consent is required since data are obtained through the National Epidemiological Surveillance approved by the Scientific Committee of the Instituto Adolfo Luiz (CTC 61-M/2020). In England, the UK Health Security Agency (UKHSA) has legal permission, provided by Regulation 3 of The Health Service Regulations 2002, to monitor the safety and effectiveness of vaccines for national surveillance of communicable diseases (http://www.legislation.gov.uk/uksi/2002/1438/regulation/3/made). In Malawi, the study protocol was approved by Malawi's National Health Sciences Research Committee (protocol 867), Kamuzu University of Health Sciences College of Medicine Research Ethics Committee (COMREC) (P-01/08/609 and P.09/09/826), and the University of Liverpool Research Ethics Committee (RETH490). Individual patient informed consent was not required for the use of publicly available anonymised routine samples as per COMREC guidelines 5·6. In South Africa, ethical approval to conduct laboratory-based and enhanced surveillance was obtained from the Health Research Ethics Committee (Human), University of Witwatersrand (M140159) and individual patient consent was obtained for clinical data collection at enhanced surveillance sites. Ethical approval for this study was granted by the London School of Hygiene and Tropical Medicine (25787).

## Reporting summary
Further information on research design is available in the Nature Portfolio Reporting Summary linked to this article.

## Data availability
The aggregated and anonymised full datasets used in this study are freely available in the GitHub via Zenodo repository[46].

## Code availability
An R script used to analyse aggregated and anonymised datasets are freely available in the GitHub via Zenodo repository[46].

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

## Acknowledgements

We thank all the individuals working in respective invasive pneumococcal disease (IPD) surveillance and laboratory management teams in the four countries for providing important data on IPD isolates and related causing serotypes. We would like to thank Stephen Gordon of Liverpool School of Tropical Medicine for insightful discussion and feedback at the initial stage of this work. D.T., K.C.J., S.F., N.F., R.S.H., and T.D.S. are supported by the National Institute for Health Research (NIHR) Global Health Research Unit on Mucosal Pathogens (MPRU) where RSH is a NIHR Senior Investigator. In addition, SF is supported by a Sir Henry Dale Fellowship jointly funded by the Wellcome Trust and the Royal Society. A project grant from the National Institute for Health and Care Research (NIHR) Global Health Research Unit on Mucosal Pathogens (MPRU) is supported using UK aid from the UK Government (Grant 16/136/46). The views expressed in this publication are those of the author(s) and not necessarily those of the NIHR or the Department of Health and Social Care. The MLW Clinical Research Programme is supported by a Strategic Award from the Wellcome, UK. This research was funded in whole, or in part, by the Wellcome Trust [Grant number 208812/Z/17/Z]. For the purpose of open access, the author has applied a CC BY public copyright licence to any Author Accepted Manuscript version arising from this submission. The funders had no role in study

design, collection, analysis, data interpretation, writing of the report or in the decision to submit the paper for publication. The corresponding author and senior authors had full access to the study data, and together, had final responsibility for the decision to submit for publication.

## Author contributions

Conceptualization—D.T., S.C., S.F., E.M., P.M., N.A.; Data curation—D.T., J.K., S.M., T.D.S., Z.A.C., N.F.Y., S.C.G.A., MCdCB. Formal analysis—D.T., S.C., S.F.; Funding acquisition—D.T., N.F., S.N.L., R.S.H., C.C., M.Cd.C.B., S.F.; Investigation—D.T., S.C., J.K., Av.G., S.W., S.M., T.D.S., E.M., P.M., N.A., Z.A.C., N.F.Y., K.C.J., N.F., S.C.G.A., S.N.L., R.S.H., C.C., M.Cd.C.B., S.F.; Methodology—D.T., S.C., N.F., S.N.L., R.S.H., C.C., M.Cd.C.B., S.F.; Project administration—J.K., Av.G., S.W., S.M., T.D.S., Z.A.C., N.F.Y., K.C.J., N.F., S.C.G.A., S.N.L., R.S.H., C.C., M.Cd.C.B.; Resources—D.T., N.F., S.N.L., R.S.H., C.C., M.Cd.C.B., S.F.; Software—D.T., S.C.; Supervision—S.F., N.F., Validation—S.C., J.K., Av.G., S.W., S.M., T.D.S., E.M., P.M., N.A., Z.A.C., N.F.Y., K.C.J., N.F., S.C.G.A., S.N.L., R.S.H., C.C., M.Cd.C.B., S.F.; Visualization—D.T., S.C.; Writing-original draft—D.T., S.C., S.F.; Writing-review & editing—D.T., S.C., J.K., Av.G., S.W., S.M., T.D.S., E.M., P.M., N.A., Z.A.C., N.F.Y., K.C.J., N.F., S.C.G.A., S.N.L., R.S.H., C.C., M.Cd.C.B., S.F.; All authors read and approved the final manuscript.

## Competing interests

The authors declare no competing interests.

## Additional information

[1]Centre for Mathematical Modelling of Infectious Diseases, London School of Hygiene & Tropical Medicine, London, UK. [2]Department of Infectious Disease Epidemiology London School of Hygiene & Tropical Medicine, London, UK. [3]Malawi Liverpool Wellcome Research Programme, Blantyre, Malawi. [4]Centre for Respiratory Diseases and Meningitis, National Institute for Communicable Diseases of the National Health Laboratory Service, Johannesburg, South Africa. [5]School of Public Health, University of the Witwatersrand, Johannesburg, South Africa. [6]School of Pathology, University of the Witwatersrand, Johannesburg, South Africa. [7]Division of Infection and Immunity, University College London, London, UK. [8]Julius Center for Health Sciences and Primary Care, University Medical Centre Utrecht, Utrecht, Netherlands. [9]University of Otago, Dunedin, New Zealand. [10]Immunisation and Countermeasures Division, UK Health Security Agency, London, UK. [11]Department of Clinical Sciences, Liverpool School of Tropical Medicine, Liverpool, UK. [12]Institute of Infection, Veterinary and Ecological Sciences, University of Liverpool, Liverpool, UK. [13]National Laboratory for Meningitis and Pneumococcal Infections, Laboratory for Meningitis, Pneumonia and Pneumococcal Infection, Centre of Bacteriology, São Paulo, Brazil. ✉e-mail: deus.thindwa@gmail.com

