## [Peer Review File · Nature Communications]

Optimal age targeting for pneumococcal vaccination in older adults; a modelling studyREVIEWER COMMENTS

Reviewer #1 (Remarks to the Author):

This is a well-reasoned and well-explained study modelling the effectiveness of a single dose of either a PCV or PPV product in older adults for the prevention of IPD. Estimates for the potential impact of PPV or PCV under a fast and slow waning condition are presented for four countries. Differences between countries are addressed well, as are the limitations in terms of surveillance data availability, particularly with regard to serotype distribution.

A few assumptions used for modelling are somewhat difficult to justify, namely the use of 100% vaccine uptake across all modelling scenarios in the older adult population, and the assumption that serotype distribution in Malawi would remain consistent across subsets of the older adult population, but both choices are appreciable given the objective of the analyses (establish maximum vaccine impact in a variety of waning, product, and population demographic scenarios) and limitations in the available data (better to assume consistent serotype distribution than to construct something unsupported and artificial).

There are a few places where the authors should work with editors to ensure consistent formatting (decimals-- midline or not? and how the journal prefers to deal with commas in lists of numbers), but the paper is well-written and well-organized.

Pneumococcal vaccine decisions like product selection and age recommendations are major public health priorities, at least in part because of the high costs of pneumococcal conjugate vaccines. The data provided by this manuscript will be useful for public health policy makers trying to navigate a wide variety of new pneumococcal vaccine products.

Reviewer #2 (Remarks to the Author):

Considering the waning of immunity, the current study aims to propose an optimal age-specific vaccination strategy under different demographic patterns, vaccine efficacies and waning scenarios. In particular, single-dose PCV/PPV vaccination strategies against invasive pneumococcal disease (IPD) risk in older adults greater than 55y in Brazil, England, Malawi, and South Africa are focused.

The manuscript is well-written and the the study is rigorously performed. However, the novel contribution of the current investigation seems limited, as similar methodologies have been widely used in other scenarios, in particular, during the ongoing COVID-19 pandemic with vaccine shortage in the early stage. Different from other studies using similar methods, the authors studied a specific disease in particular countries/regions. The authors may like to discuss related studies to attract a wider readership.

Another main concern is the population demographic diagrams in Figure 1 for England and Malawi, which are not monotonically decreasing. Theoretically, these patterns are not stable. As a result, the optimal ages in Figure 2, which are consistent with those demographic patterns in Figure 1, are not always 55. It is expected that if the age-stratified population is stabilized at a steady state, the optimal age for pneumococcal vaccination would always be 55, as predicted for Brazil, Malawi and South Africa.

One minor comment: More details on incorporating vaccine efficacy in the model should be added in the METHODS part, in particular, on reducing the expected incidence.

Reviewer #3 (Remarks to the Author):

This manuscript by Thindwa et al estimates the impact of hypothetical adult pneumococcal vaccination programs against invasive pneumococcal disease (IPD) in different settings (LIC, MIC, and HIC) with different IPD epidemiology and population demographics. Their model is not too

complex, yet considers key factors (e.g., time since pediatric PCV implementation, accounting for PPSV23 use in the England, vaccine waning pattern, differences in HIV prevalence) and an important uncertainty (e.g., differences in VE by age). They also present the findings as number of cases prevented in a lifetime, and as number of individuals needed to vaccinate. The manuscript is well-written, and data from this study will help inform global pneumococcal vaccination policy for adults.

I do have some questions/comments for clarification.

- Lines 54-57: it was not clear to me what these percentages are representing. Are these the proportion of IPD preventable in a lifetime (i.e., denominator is all IPD that can occur in a cohort of adults 55 years and older, and the numerator is the number of IPD cases that can be prevented if vaccinated at a certain age)? Same comment for the results section starting from line 131, "Optimal age for vaccination". Suggest defining the % in the methods.

- Figure 1 right panel: I notice the striking differences in the Y axis scale between England/Malawi vs Brazil/South Africa (and I assume that is resulting in the very high number needed to vaccinate in these countries). Do we have any concerns about the accuracy of IPD case ascertainment among older adults in these countries?

- Line 222: I think this is the first time PCV21 is introduced in this manuscript. Please consider providing some explanation (i.e., investigational 21-valent pneumococcal conjugate vaccine). Also, PCV21 covers 8 additional serotypes that are not contained in currently available vaccines, so I am not sure if that could be called "most serotypes not targeted by PPSV23"

<https://www.merck.com/news/merck-announces-u-s-fda-has-granted-breakthrough-therapy-designation-for-v116-the-companys-investigational-21-valent-pneumococcal-conjugate-vaccine-for-the-prevention-of-invasive-pneumococ/>

- Supplemental Table S3—please check the title of the table. It does not seem to reflect the content of the table?

Reviewer #1:

1. *This is a well-reasoned and well-explained study modelling the effectiveness of a single dose of either a PCV or PPV product in older adults for the prevention of IPD. Estimates for the potential impact of PPV or PCV under a fast and slow waning condition are presented for four countries. Differences between countries are addressed well, as are the limitations in terms of surveillance data availability, particularly with regard to serotype distribution.*

Response: Thank you.

2. *A few assumptions used for modelling are somewhat difficult to justify, namely the use of 100% vaccine uptake across all modelling scenarios in the older adult population, and the assumption that serotype distribution in Malawi would remain consistent across subsets of the older adult population, but both choices are appreciable given the objective of the analyses (establish maximum vaccine impact in a variety of waning, product, and population demographic scenarios) and limitations in the available data (better to assume consistent serotype distribution than to construct something unsupported and artificial).*

Response: Thank you for the comment. We have added text to “Discussion” section to clarify on implication of lower vaccine coverage in this model. “Although our modelled scenarios assumed 100% vaccination coverage, which may not reflect real-world vaccine uptake, we aimed to characterise the potentially preventable burden and that the predicted impact will scale linearly with uptake in this model thus not affecting optimal age of vaccination unless uptake is differential by age groups.

(Lines 198-202).” Indeed, the assumption of serotype distribution in Malawi is the best we could use in the presence of limited data.

3. *There are a few places where the authors should work with editors to ensure consistent formatting (decimals-- midline or not? and how the journal prefers to deal with commas in lists of numbers), but the paper is well-written and well-organized.*

Response: These have been rectified.

4. *Pneumococcal vaccine decisions like product selection and age recommendations are major public health priorities, at least in part because of the high costs of pneumococcal conjugate vaccines. The data provided by this manuscript will be useful for public health policy makers trying to navigate a wide variety of new pneumococcal vaccine products.*

Response: We totally agree.

Reviewer #2:

1. *Considering the waning of immunity, the current study aims to propose an optimal age-specific vaccination strategy under different demographic patterns, vaccine efficacies and waning scenarios. In particular, single-dose PCV/PPV vaccination strategies against invasive pneumococcal disease (IPD) risk in older adults greater than 55y in Brazil, England, Malawi, and South Africa are focused.*

Response: Thank you.

2. *The manuscript is well-written, and the study is rigorously performed. However, the novel contribution of the current investigation seems limited, as similar methodologies have been widely used in other scenarios, in particular, during the ongoing COVID-19 pandemic with vaccine shortage in the early stage. Different from other studies using similar methods, the authors studied a specific disease in particular countries/regions. The authors may like to discuss related studies to attract a wider readership.*

Responses: The manuscript addresses an important public health question of considerable interest to clinicians, epidemiologists, vaccinologists and policymakers (including the WHO SAGE pneumococcal vaccine working group). Pneumococcal conjugate vaccines (PCVs) are highly effective in preventing vaccine-type pneumococcal disease in children yet whether and when to deploy PCVs in adults remains uncertain. To address this, we have used an existing analytic method to identify the optimal age-targeting for a single-dose pneumococcal vaccination against VT-IPD in older adults that considers key factors including population demographics, IPD incidence, vaccine efficacy or effectiveness, and vaccine efficacy waning. We only know of a single related study that has assessed the role of timeliness in cost-effectiveness of pneumococcal adult vaccination, which was in the Australian context (Chen C et al.). We have not modified the text further as we are uncertain that a discussion of the application of our approach to other vaccine targets is central to the important messages in our paper and are concerned that to do so would considerably increase the word count.

3. *Another main concern is the population demographic diagrams in Figure 1 for England and Malawi, which are not monotonically decreasing. Theoretically, these patterns are*

not stable. As a result, the optimal ages in Figure 2, which are consistent with those demographic patterns in Figure 1, are not always 55. It is expected that if the age-stratified population is stabilized at a steady state, the optimal age for pneumococcal vaccination would always be 55, as predicted for Brazil, Malawi and South Africa.

Response: To partly address the problem of demographic instability, we smoothed annual age population estimates in all countries using a 5-years moving average as explained in the “Methods” section (lines 286-293). Since optimal vaccine impact is partly dependent on age-specific IPD incidence, it is expected that in a long-term, each country monitors changes in their population trend to make updated predictions of optimal age targeting. Thus, we have added text in the “Discussion” section to explain impact of demographic changes. “It is also worth noting that our study uses demographic snapshots and that there are ongoing demographic changes, particularly in Africa. Generally, with increasing life expectancy we would expect the optimal age of vaccination to increase, but this is a process spanning decades.” (line 251-254)

4. *One minor comment: More details on incorporating vaccine efficacy in the model should be added in the METHODS part, in particular, on reducing the expected incidence.*

Response: We have added text on the cohort model. “Vaccine impact in the cohort model was a function of 1) smoothed population counts, 2) fitted IPD incidence using exponential growth model, and 3) fitted vaccine VE using piecewise constant model, stratified by age, country, vaccine product, age-dependency on initial VE, and waning VE assumption.” (lines 361-365)

Reviewer #3:

1. *This manuscript by Thindwa et al estimates the impact of hypothetical adult pneumococcal vaccination programs against invasive pneumococcal disease (IPD) in different settings (LIC, MIC, and HIC) with different IPD epidemiology and population demographics. Their model is not too complex, yet considers key factors (e.g., time since pediatric PCV implementation, accounting for PPSV23 use in the England, vaccine waning pattern, differences in HIV prevalence) and an important uncertainty (e.g., differences in VE by age). They also present the findings as number of cases prevented in a lifetime, and as number of individuals needed to vaccinate. The manuscript is well-written, and data from this study will help inform global pneumococcal vaccination policy for adults.*

Response: Thank you.

2. *Lines 54-57: it was not clear to me what these percentages are representing. Are these the proportion of IPD preventable in a lifetime (i.e., denominator is all IPD that can occur in a cohort of adults 55 years and older, and the numerator is the number of IPD cases that can be prevented if vaccinated at a certain age)? Same comment for the results section starting from line 131, “Optimal age for vaccination”. Suggest defining the % in the methods.*

Response: Thank you for the comment. The percentage definition has now been added in the “Methods” section. “In particular, we calculated the percentage of preventable cases as the proportion of IPD preventable in a lifetime, with denominator being all IPD that

can occur in a cohort of adults ≥ 55 y, and the numerator as the number of IPD cases that can be prevented if vaccinated at a certain age cohort.” (lines 370-373)

3. *Figure 1 right panel: I notice the striking differences in the Y axis scale between England/Malawi vs Brazil/South Africa (and I assume that is resulting in the very high number needed to vaccinate in these countries). Do we have any concerns about the accuracy of IPD case ascertainment among older adults in these countries?*

Response: There is some expected degree of underreported cases in low/middle-income countries partly due to limited resources. However, we found it reasonable to assume that underreporting is consistent across adult age groups, such that a relative change in IPD incidence by age to identify optimal age-targeting vaccination is less likely affected. This has been explained in the “Discussion” section (lines 243-249).

4. *Line 222: I think this is the first time PCV21 is introduced in this manuscript. Please consider providing some explanation (i.e., investigational 21-valent pneumococcal conjugate vaccine). Also, PCV21 covers 8 additional serotypes that are not contained in currently available vaccines, so I am not sure if that could be called “most serotypes not targeted by PPSV23 ” <https://www.merck.com/news/merck-announces-u-s-fda-has-granted-breakthrough-therapy-designation-for-v116-the-companys-investigational-21-valent-pneumococcal-conjugate-vaccine-for-the-prevention-of-invasive-pneumococ/>*

Response: The sentence has been modified to include more information about PCV21.

“On the other hand, phase 3 trials of the investigational PCV21 by MERCK are underway after receiving a breakthrough therapy designation from the US Food and Drug

Administration for prevention of IPD. PCV21 targets serotypes that account for 85% of IPD in >65y in the US including 8 serotypes not targeted by currently licensed vaccines, and is it expected to improve tackling the remaining burden of pneumococcal disease in older adults” (lines 216-223)

5. *Supplemental Table S3—please check the title of the table. It does not seem to reflect the content of the table.*

Response: This has been corrected. “Table S3. A base case scenario, age-independent initial efficacy/effectiveness (VE), of PPV23 or PCV20 use in 55- and 70- years old cohort. Reduction in vaccine-type IPD cases between fast vs slow waning VE relative to no vaccination scenario.”

Editorial Note: The authors respond below to comments left by Reviewer #1 in a tracked changes copy of the manuscript

Miscellaneous comments in the manuscript:

1. *Suggest removing either "however" or "yet" (Line 71)*

Response: This has been corrected. “However, only a single study has assessed the age at which the most gain from such programme is seen (in the Australian context)”

2. *Midline decimals are not usually used in this journal, check formatting guidelines*

Response: These have been rectified.

3. *Insert an “and” here so it doesn't look like 19,537 cases (line 88)*

Response: This has now been added. “*During the study period, Brazil (2015-2017), England (2016-2019), Malawi (2016-2019) and South Africa (2015-2018) reported 559, 13,971, 19, and 537 IPD cases.*”

4. *Also, In the base scenario with rapid waning VE, higher proportion of all preventable cases are estimated for using PCV20 vs PPV23 (lines 129)*

Response: “a” has been added between the underlined words. “Also, in the base scenario with rapid waning VE, a higher proportion of all preventable cases are estimated for using PCV20 vs PPV23”

5. *Furthermore, higher proportion of IPD cases are preventable under slow vs rapid waning of PPV23 VE among adults aged 55y. How are these (slow vs rapid waning) defined? (line 133)*

Response: “a” has been added between the underlined words (line 133). We have also added text in the “Methods” section to describe slow and fast waning of vaccine efficacy: “We considered waning VE from Andrews et al. and Djennad et al. in the vaccination impact cohort model as they provided the fastest and slowest waning VE, respectively, of the four studies. The piecewise constant functions that define initial VE and waning VE are shown in supplementary (Fig S2, Fig S3).” (lines 343-346).

6. *If a scenario of age-dependent initial VE is considered in a sensitivity analysis, vaccine impact (total cases averted) remains maximum at 55y in Brazil, Malawi, and South Africa*

irrespective of vaccine product and assumption of waning VE as was the case with the base case scenario of age-independent initial VE (line 166).

Response: The underlined word has been corrected to “highest”.

7. *The increasing incidence of IPD in older adulthood is outweighed by many more individuals in the fifties in LIC/MICs. (line 185)*

Response: The underlined word has been changed. “The increasing incidence of IPD in older adulthood is outweighed by many more individuals in their fifties in LIC/MICs.”

8. *Did you do any sensitivity analyses to reflect lower levels of uptake? I understand that this is defining a maximum possible impact, but it would be nice to see the impact of 70% or 50% uptake, since these might be more programmatically realistic*

Response: We have added text to Discussion section to clarify on implication of reduce vaccine coverage in this model. “Although our modelled scenarios assumed 100% vaccination coverage, which may not reflect real-world vaccine uptake, we aimed to characterise the potentially preventable burden and that the predicted impact will scale linearly with uptake in this model thus not affecting optimal age of vaccination unless uptake is differential by age groups. (Lines 198-202).”